# The Use of Additives to Prevent Urolithiasis in Lambs Fed Diets with a High Proportion of Concentrate

**DOI:** 10.3390/vetsci10100617

**Published:** 2023-10-11

**Authors:** Vicente Luiz Macêdo Buarque, Helena Viel Alves Bezerra, Guilherme Pegoraro Rissi, Regner Ítalo Gonçalves de Oliveira, Natália Marques da Silva, Nara Regina Brandão Cônsolo, Germán Darío Ramírez-Zamudio, Ricardo de Francisco Strefezzi, Sarita Bonagurio Gallo, Saulo Luz Silva, Paulo Roberto Leme

**Affiliations:** 1Department of Animal Science, College of Animal Science and Food Engineering, University of São Paulo (USP), Pirassununga 13635-900, SP, Brazil; vicentebuarque@usp.br (V.L.M.B.); helena.bezerra@usp.br (H.V.A.B.); guilherme.rissi@usp.br (G.P.R.); regneritalo@usp.br (R.Í.G.d.O.); natalia.marques.silva@usp.br (N.M.d.S.); saritabgallo@usp.br (S.B.G.); sauloluz@usp.br (S.L.S.); prleme@usp.br (P.R.L.); 2Department of Animal Nutrition and Production, School of Veterinary Medicine and Animal Science, University of São Paulo (USP), Pirassununga 13635-900, SP, Brazil; nara.consolo@usp.br; 3Department of Veterinary Medicine, College of Animal Science and Food Engineering, University of São Paulo (USP), Pirassununga 13635-900, SP, Brazil; rstrefezzi@usp.br

**Keywords:** ammonium chloride, benzoic acid, calcium chloride

## Abstract

**Simple Summary:**

Urolithiasis is the formation of salt crystals in the kidney and can be caused by several factors, one of which is the use of unbalanced diets for feeding animals. This study aims to evaluate the effectiveness of different additives in preventing urolithiasis in lambs fed a high-concentrate diet. Thirty-two noncastrated male lambs were divided into four treatments, three including additives in the diet and one group without additives (CON), and fed for 56 days in a feedlot. The additives used were ammonium chloride (ACL), calcium chloride (CCL), and benzoic acid (BZA), and each was added to a diet composed of 6% *Cynodon* ssp. hay and 94% concentrate. The use of additives in the diet did not affect the performance of the lambs. Regarding urine pH, inclusion of ACL and CCL in the diet acidified the urine, but none of the additives prevented urolithiasis. However, the formation of these crystals did not show obstruction of the urinary tract or affect the lambs’ health.

**Abstract:**

This study aimed to evaluate the effectiveness of different additives in preventing urolithiasis in lambs fed a diet rich in concentrate and their impact on performance and blood and urinary parameters. Thirty-two noncastrated male lambs, crossbred Dorper × Santa Inês, with initial body weights (BWs) of 23 ± 0.1 kg and ages of 50 ± 5 days, were kept in individual pens and fed a diet composed of 6% *Cynodon* ssp. hay and 94% concentrate and subjected to four treatments: CON without inclusion of additives, addition of ACL 5 g/kg of dry matter (DM), addition of CCL 6.3 g/kg of DM, and addition of BZA 5 g/kg of DM. There was no effect of treatment or interaction with time on blood parameters (*p* > 0.050), and performance characteristics, morphometry of ruminal papillae, and scores of cecum and rumen lesions were not affected by the addition of additives to the diet (*p* > 0.050). Greater urinary acidification was observed in animals from CCL and ACL treatments (*p* = 0.033). Calcium chloride acidified the urinary pH and can be used instead of ammonium chloride at a concentration of 0.63% based on DM, when this is the objective.

## 1. Introduction

With constant population growth, an increase of 2.2 billion people is estimated by the year 2050, according to data from the United Nations [1]. Consequently, it is necessary to intensify livestock production, including lamb farming. Animals finished in a feedlot with high-concentrate diets is the leading way to intensify the system production. Nevertheless, this strategy can increase the occurrence of diseases of nutritional or metabolic origin, such as urolithiasis [2,3].

Urolithiasis is the formation of crystals of mineral salts, which can partially or entirely obstruct the urinary tract [4], leading to severe consequences, such as decreased performance or even death of animals. This is a complex etiology disease [5] and is frequently observed in sheep raised in intensive production systems [6]. Several factors can contribute to the formation of uroliths: nutritional, caused by imbalanced diet and the hardness of the water consumed by ruminants [7], and physiological factors, due to the anatomy of the urethra, sex, age, breed, and water restriction, in addition to the season of the year and the geographic region [8].

According to Riet-Correa et al. [9], significant incidence occurs in males due to the anatomy of the urinary tract, consisting of the sigmoid flexure, ischial curvature, and urethral process, which are the primary sites for the deposition of uroliths, resulting in the obstruction of the urinary tract. After clinical signs of the disease, animals hardly recover and generally become unable to reproduce and are discarded [10].

In sheep and goats, the most common uroliths are composed of calcium carbonate, followed by struvite (ammonium and magnesium phosphate). The formation of these crystals occurs at a pH ranging between 7.2 and 8.8 and its dissolution at a pH of less than 6.5. Other crystals, such as apatite (calcium phosphate), occur at a pH intermediate between 6.6 and 7.8. Thus, urinary pH plays a fundamental role in developing calcium carbonates, apatite, and struvite uroliths [11], proving its importance for the urinary concentration of these calculogenic compounds. According to Goff [12], anionic diets induce metabolic acidosis through the compensatory increase in extracellular hydrogen ions (H+), where the kidneys excrete the H+ excess to maintain electroneutrality, producing more acidic urine.

The NRC recommends preventive nutritional strategies using acidifiers [6], such as including 0.5% ammonium sulfate in the diet or using ammonium chloride at 100 to 200 mg/kg of body weight. Although ammonium chloride is the most used preventive method due to its acidifying capacity, a high inclusion of this molecule can reduce feed intake because it is not palatable [13].

Thus, there is a need to explore alternative additives to ammonium chloride that can acidify the urine of animals and prevent the development of urolithiasis. Among some available additives, calcium chloride and benzoic acid can be potential substitutes for ammonium chloride. Unlike calcium chloride, benzoic acid does not increase the anionic content of the animal’s diet; instead, it is metabolized by the liver to form hippuric acid, which is excreted in the urine to lower its pH. Thus, it was hypothesized that, due to their acidifying capacity, these additives are effective in preventing obstructive urolithiasis, through the acidification of urinary pH, without depressing dry matter intake and animal performance. Therefore, this study aimed to evaluate the effectiveness of different additives in preventing urolithiasis in lambs fed a high-concentrate diet and their impact on performance, blood, and urinary parameters.

## 2. Materials and Methods

All procedures used in this study were conducted in accordance with the Institutional Animal Care and Use Committee Guidelines (protocol 4633210120) approved by the committees of the Faculty of Animal Science and Food Engineering, University of São Paulo.

### 2.1. Animals, Facilities, and Treatments

The experiment was conducted at the experimental facility of the Laboratory of Animal and Meat Quality Assessment of the Department of Animal Science, Faculty of Animal Science and Food Engineering (FZEA/USP), Pirassununga–SP, Brazil (21°59′ south latitude and 47°26′ west longitude).

Thirty-two male lambs, noncastrated, crossed Dorper × Santa Inês, with initial BWs of 23 ± 5 kg and ages of 50 ± 5 days were housed in individual pens (2 × 1.25 m), with ad libitum access to feed and water. The animals were adapted to the facilities and diet for 10 days, with the supply of a control diet (CON), without the inclusion of additives. Afterwards, the animals were assigned to a random block design according to their initial BWs into four treatments, with eight replications per treatment: CON, without additives; ACL, inclusion of 5.0 g/kg DM of ammonium chloride; CCL, inclusion of 6.3 g/kg DM of calcium chloride; BZA, inclusion of 5.0 g/kg DM of benzoic acid. The diet with each additive was mixed for 10 min in a horizontal mixer (Máquinas Pereira Londrina–Mapelon, Londrina, Paraná, Brazil).

The diet was composed of 6% roughage *Cynodon* sp. hay and 94% concentrate containing ground corn, soybean meal, urea, and minerals (Table 1), and was formulated according to the recommendations of the NRC [6] for finishing lambs, differing only in the inclusion or not of the additive in relation to the control treatment and provided for 56 days.

The DM intake (DMI) was calculated daily and adjusted considering a margin of 10% orts in natural matter offered, allowing for ad libitum consumption. The animals were weighed, without fasting, at the beginning and every 14 days throughout the experimental period until day 56. The average daily gain (ADG) was calculated at the end of the feeding period as the coefficient of the estimated regression of the body weight as a function of the feedlot time.

### 2.2. Sampling and Chemical Analysis

Samples of the total diet were collected every week and stored in appropriate plastic bags for the analysis of dry matter, crude protein, ether extract, and acid detergent fiber according to the methods of AOAC [15]. Neutral detergent fiber was analyzed using α-amylase [16] without the addition of sodium sulfite. Total digestible nutrients were estimated according to Weiss et al. [14].

For the analysis of serum biochemistry and blood electrolytes, 5 mL of blood samples were collected on the day prior to the start of the experiment and at 28 and 56 days prior to the food offer, through puncture of the jugular vein in vacuum tubes with clot activator gel. The urea and creatinine concentrations in these blood samples were determined using commercial kits (Labtest, Lagoa Santa, Minas Gerais, Brazil) through the following methods: enzymatic–colorimetric and kinetic–colorimetric, respectively, with readings performed on a spectrophotometer. For the hemogram, samples of 5 mL of blood were collected through venipuncture in vacuum tubes containing EDTA and processed by the method of automatic cell counting, using a Mindray BC-2800 Vet device (Mindray, Gurugram, Haryana, India).

The urine collections were carried out at 7:00 a.m. before feeding with the animals individually contained in station and the samples obtained through natural urination or stimulation of the foreskin to obtain 5 mL of urine. Three urine collections were carried out, before treatments and on 28 and 56 days of feedlot. The pH of each sample was measured using a digital pH meter, model HI 99163 (Hanna Instruments, São Paulo, Brazil) and sent to the Clinical Laboratory of the Hospital Clinical Unit (UDCH) to perform the urinalysis test according to the methodology of Kaneko et al. [17].

### 2.3. Slaughter and Sample Collection

After 56 days on feed, the lambs were transported to the experimental slaughterhouse at the University of Sao Paulo, 200 m from the barn. The slaughterhouse is authorized by the Inspection Service of the State of São Paulo (SISP) and follows the humanitarian procedures established by Brazilian legislation (Decree 39,972 of 17 February 1995), with stunning of the animals by a penetrating captive bolt and bleeding through the jugular vein and carotid artery. The carcasses were then skinned, eviscerated, washed, identified, and weighed.

System organs (kidney, ureter, bladder, penile urethra, and urethral process) were collected and stored for 5 µm histological sections in a microtome (RM2255-Leica Biosystems, Leica do Brasil Importação e Comércio Ltda, São Paulo, Brazil) and the preparation of slides, which were stained with hematoxylin and eosin (HE), and the alterations were analyzed according to the methodology described by Pluske et al. [18].

The rumenitis score was performed for the observation of lesions to the rumen epithelium classified on a scale of 0 to 10, where each score represents 10% of the injured rumen [19]. Fragments of 1 cm^2^ of the cranial sac of the rumen were used to determine the average number of ruminal papillae counted by three persons, and the mean of the three counts was considered, as well as the papillary surface absorption in percentage, the average area of ruminal papillae in square centimeters, and the absorptive surface area for absorption per cm^2^ of wall using the image analysis program UTHSCSA Image Tool [20]. The cecal epithelium was classified according to the presence of petechiae using a lesion scale of 0 to 10.

### 2.4. Statistical Analysis

Plots of residuals and the W statistic [21] were evaluated to determine normality for all data, and outliers (>3 and <−3 standard deviation) were excluded. Data were analyzed in a completely randomized design (initial body weight, *n* = 2), with four treatments and eight replications per treatment, using a mixed procedure of SAS 9.4 (SAS Institute Inc., Cary, NC, USA) with 32 animals distributed in four treatments, totaling eight replicates per treatment.

For the performance, papillary morphometry, and cecal pH data and physicochemical parameters of urine, treatment was used as a fixed effect and the block (initial body weight) as a random effect. Traits evaluated over time (urine pH, blood, and serum traits) were analyzed as time repeated measurements, considering treatment, days of measurement, and its interaction as fixed effects and the random effect of block and animal (subject). Residual covariance structures were modeled, and the best-fitted one, based on the Bayesian information criterion (BIC), was used. The significance was declared at *p* ≤ 0.05, when a significant effect was observed, and means were compared by the Student’s *t*-test.

Ruminitis scores and cecal lesions did not show a Gaussian distribution and were analyzed using the SAS GLIMMIX procedure using a Poisson distribution.

## 3. Results

### 3.1. Feed Intake and Performance

There was no effect of treatment on performance characteristics (*p* > 0.05; Table 2). The DMI % BW ranged from 3.33% to 3.53%, corresponding to 1.167 to 1.244 kg daily DMI. The ADG ranged from 0.333 to 0.357 kg, and the average feed efficiency was 283 g of weight gain per kg of DM ingested. The average final weight was 42.65 kg between treatments.

### 3.2. pH and Physical–Chemical Parameters of Urine

There was a significant interaction between treatment and time (*p* = 0.033; Figure 1) and treatment (*p* < 0.001) for urine pH. Urinary pH in CON and BZA treatments did not differ from each other and presented values higher than 7.0. Animals fed the CCL diet had the lowest mean urinary pH value and differed from the others, followed by ACL with a pH value around 7.0 and linearly reduced over time (*p* = 0.001).

On day 28, there was an significant reduction in the mean value of urinary pH in the CCL treatment, which remained lower during the following days compared with the other treatments (Figure 1). The pH values lower than 7.0 were observed only in the treatments CCL and ACL at the end of the experiment. Animals fed CON and BZA diets did not present significant changes in urinary pH values throughout the experimental period (*p* > 0.05), maintaining constant pH values above 7.0.

The odor in all urine samples was classified as sui generis, and the presence of glucose, ketone bodies, cylinders, crystals, or mucus was not detected (Table 3). The urobilinogen was within the normal limits, and in general, there was no presence of red blood cells except in nine animals. However, the small amount makes this finding without clinical significance, according to Santarosa et al. [22].

There was a treatment effect for color, density, and protein of the urine (*p* = 0.001; *p* = 0.004; *p* = 0.006, respectively). However, treatments did not affect the aspect of turbidity (*p* = 0.148) and the presence of blood (*p* = 0.189). A urinary concentration greater than 1.040 was found in 13.33% of the samples in the BZA treatment, CON corresponded to 10.00% of the total, and in the CCL and ACL treatments, the density greater than 1.040 corresponded to 6.67% and 1.11%, respectively, of the total of analyzed samples.

In general, about 64% of the samples evaluated were negative or had trace amounts of protein, while approximately 36% of the total samples had at least 30 mg/dL of protein. However, of this 36%, only 34% had a concentration greater than 100 mg/dL, observed more frequently in animals fed with the BZA diet than those with the other treatments. In the kidney, urinary bladder, urethra, and urethral process, evaluated by histopathological analysis, no noteworthy alterations to histopathology were observed, presenting typical and preserved structures, without the presence or evidence of the lithogenesis process, regardless of the treatment to which the animals were submitted.

However, the results obtained from the ureter samples showed intense luminal stenosis, which is the narrowing of the ureter canal, with modification of the epithelium, intense fibrosis, and thickening of the lamina propria (Figure 2). Furthermore, in all treatments, the samples contained calcified uroliths occupying the organ’s lumen, indicating that the additives did not have the expected preventive effect.

### 3.3. Morphological and Health Parameters in the Rumen and Cecum

There was no effect of treatment on the papillae morphometric characteristics (*p* > 0.05; Table 4). However, treatments changed the cecal pH (*p* = 0.034; Table 4), with higher and similar values for animals fed ACL and CON, which differed from the CCL and BZA groups. However, it is possible to observe that there was no damage related to the ruminal and cecal health of the lambs, regardless of the treatment used.

There was no effect of treatments on the rumenitis score (*p* = 0.247; Figure 3). The highest frequency of animals with a rumenitis score of less than three was observed in all treatments, with a score of zero having greater occurrence in animals fed the ACL diet. Despite this, the highest score of grade seven was observed in animals fed the ACL diet, being the only one among all animals to present such a degree of damage to the rumen epithelium. Grade four lesions were found in the rumen of only two animals from the CCL and BZA treatments. This degree of rumen damage was not observed in the other treatments.

There were no differences in cecal lesion scores between treatments (*p* = 0.302; Figure 4). However, 12.5% of the animals in each of the CCL and BZA treatments and 6.25% in each of the CON and ACL groups had a cecal lesion score of 10.

### 3.4. Blood Parameters

Red blood cells, hematocrit, mean corpuscular volume, mean corpuscular hemoglobin, and mean corpuscular hemoglobin concentration were not influenced by the treatments or the interaction between treatment and time (*p* > 0.05). However, diet affected hemoglobin (*p* = 0.043; Table 5), with approximately 1 g/dL more in the CON and BZA groups than in the ACL and CCL treatments. Additionally, time reduced the concentrations of HCM and CHCM (*p* = 0.013; *p* = 0.024, respectively).

There was no effect of treatments on leukocyte indexes (*p* > 0.05; Table 6). On the other hand, there was a significant time effect on total leukocytes, segmented neutrophils, and lymphocytes (*p* = 0.044; *p* = 0.024; *p* = 0.007, respectively). Total leukocytes/µL were reduced from the 1st to the 28th day on feed, from 7431 to 6516, respectively. However, on the 56th day of feeding, the concentration of leukocytes was similar to the beginning of the dietary treatment (7856 leukocytes/µL). From the 28th day on feed, there was a 25% reduction in the concentration of segmented neutrophils, and these remained low until the 56th day on feed. In comparison, the concentration of lymphocytes increased by 30% from day 28 to 56 on feed. Nevertheless, the values obtained occurred within the variations found in the literature [23], except for eosinophils and basophils.

There was no effect of the treatments or the interaction between treatment and time for serum biochemistry (*p* > 0.05; Table 7). However, serum concentrations remained within the normal range for the specie, except for creatinine and phosphorus, with values outside the reference ranges reported by Meyer and Harvey [23]. There was a time effect for almost all serum biochemistry compounds (*p* < 0.001; Table 7), except for platelets, chloride, and potassium (*p* = 0.248; *p* = 0.200; and *p* = 0.249, respectively). In general, the serum concentrations of the compounds analyzed in this test increased with days on feed. However, there was a reduction in creatinine on day 28 on feed (*p* < 0.001), with concentrations below the reference range of 1.2 to 1.9 mg/dL in all periods, while phosphorus was higher than the range of 5.0 to 7.3 mg/dL proposed by Meyer and Harvey [23].

## 4. Discussion

Studies comparing ACL with BZA or CCL in lamb or sheep diets in the same assay have yet to be reported in the literature. However, some trials with these additives in sheep or other livestock species show variable results on productive performance. For example, Ferreira et al. [24] did not observe differences in sheep-fed ACL performance compared with the control group. Likewise, studies with crossbred steers fed a high-grain diet and an inclusion of 0.5% BZA compared with a yeast (*S. cerevisiae*) treatment [25], or in order to replace monensin and tylosin [26], did not observe differences in animal performance. However, pigs supplemented with 0.5% BZA in the diet showed higher DMI and better performance when compared with the control group, which can be due to changes in the population of gastric bacteria, such as lactic acid bacteria and *E. coli* [27]. These differences in response to benzoic acid supplementation on DMI and performance between ruminants and monogastric may be associated with the biological model in which this additive was applied.

Animals fed with CCL diet presented lower urinary pH on day 28, and on day 56, CCL and ACL treatments presented lower pH. In other words, with the objective of acidifying the urine, CCL can replace ACL. Animals fed BZA diet obtained a pH value similar to those fed ACL and CON diet, and may eventually not be as effective for this purpose. Data from the present study corroborate with those of Marques et al. [28], who provided 250 g/animal of ACL daily and observed a reduction in urinary pH from 7.89 to 5.17, while the control treatment, without anionic diet, presented a pH greater than 6.5; thus, the authors highlighted the importance of renal function in the acid–base balance. Different results were reported by Gomide et al. [29], who, when manipulating the dietary cation–anion difference (DCAD) using ACL as an anion source, did not observe differences in the urinary pH of lambs, with pH values of 7.38, 7.06, 7.36, and 8.39 for the values of DCAD of −12, +30, +76, and +133, respectively.

There was no difference in the morphometry of ruminal papillae, regardless of the treatment, without significant damage to the rumen epithelium in the function of the treatment compared with the CON group. Likewise, there was no significant difference for the papillary surface absorption, average number of ruminal papillae, and absorptive surface area. In the present study, these results between treatments may have occurred due to the similarity between diets. Xu et al. [30] compared a control diet with 94.6% hay with a diet composed of 60% concentrate for 4 weeks and found a significant increase in morphometric variables with the supply of high-concentrate diets; this increase is considered a process of adaptation of the ruminal epithelium to greater production and absorption of short-chain fatty acids (SCFA).

There was a treatment effect on cecal pH, with the highest value of pH being observed in the animals fed with the ACL diet. Little is known about the cecal metabolism in lambs; however, considering that the cecum and colon are responsible for about 8.6% to 16.8% of the total SCFA production [31], with the data found, it is possible to infer that the cecal acidification caused by acidifiers may favor the absorption of these fatty acids in this organ. In a study carried out in the 1960s, Myers et al. [32] observed that cecal pH exerts influence on the mechanisms of absorption of SCFA, and verified a reduction in absorption when there was an increase in pH from 6.2 to 7.5.

In a study evaluating cottonseed meal supplementation, Krysl et al. [33] observed that this supplementation altered ruminal fluid dynamics, but had minimal effects on cecal dynamics, with no changes in fermentation measurements and cecal pH in sheep. In the erythrogram analysis, the treatment affected only the hemoglobin concentration, with the highest concentration observed in the animals treated with BZA. When using vitamin C as an acidifier, the mean hemoglobin values found by Maciel et al. [34] were similar to those in the current study, 12.73 and 12.77 g/dL, respectively, both within the reference range proposed by Kramer [35] of 9 to 15 g/dL.

Treatments did not influence leukocyte concentrations. However, there was an effect of time on total leukocytes, neutrophils, and lymphocytes, which, although they were smaller than those found by Maciel et al. [34], remained within the reference range between 4000 and 12,000 /µL proposed by Meyer and Harvey [23]. There was no effect of treatment on serum biochemistry. However, serum component concentrations, except platelets, chloride, and potassium, increased with time. Despite this increase in serum biochemical components as a function of days on feed, it was insufficient to cause azotemia corroborated with values within the reference range [17,36].

The serum phosphorus levels found in this study ranged between 8.38 and 8.93 mg/dL, within the range proposed by the NRC [6], which is 6 to 9 mg/dL, as well as the results reported by Souza et al. [37], who determined values between 4.06 and 12.7 mg/dL for sheep in the age range of the animals in the present study. These values are higher than the reference values of 1.2 to 2 mg/dL [17] and 1.2 to 1.9 mg/dL [23]; however, the possible nutritional variations [7], breed, sex, and even the climatic characteristics [8] must be considered, as well as where the studies are carried out.

The linear increase in serum phosphorus as a function of time may have occurred due to the supply of diets with a high proportion of concentrate. Louvandini and Vitti [38], in a metabolic assay, provided different daily levels of phosphorus for sheep (1, 2, and 3 g/animal) and verified a relation of r = 0.90 between the phosphorus absorbed and the phosphorus consumed. According to Minervino et al. [39], the greater presence of phosphorus in the diet reflects a higher blood concentration. Thus, the phosphorus in the dietary ingredients, especially corn, may have caused this increase in addition to time.

Treatment affected urinary density, which might help interpret renal functions, considering that the kidneys are the organs responsible for regulating, excreting, and performing endocrine functions [40]. In all treatments, the highest frequency of urine was observed with density values below or within the reference range for the species ranging from 1015 to 1045 [41,42], except for animals fed with the BZA diet, which presented inverse action, with greater frequency of animals with lower urinary concentration. In addition, animals treated with ACL had a higher frequency of urine with a density < 1.015, which may result from increased diuresis, as reported by Jones et al. [43], who associated this effect with ACL.

The histopathological results showed that the treatments did not prevent the formation of uroliths in the ureters, in which all animals from all treatments had calculi in the lumen. Even in animals treated with BZA and ACL, with more acidic urinary pH, below 6.6, uroliths were observed. Likewise, the animals from the BZA and CON treatments showed a similar condition with a urinary pH above 7.0. This is probably due to the diet composition used in this study with a high proportion of concentrate, in which, according to Riet-Correa et al. [9], Antonelli et al. [2], and Guimarães et al. [3], the use of this type of diet can lead to the development of urolithiasis.

Santarosa et al. [22] reported that the presence of crystals indicates tubular aggression due to their formation. For the authors, this could be associated with ACL ingestion. However, in that study, as it was only observed in one animal, this hypothesis was not confirmed; for that, it would be necessary to have a more significant number of animals with tubular degeneration and necrosis, which did not occur.

Despite the observation of uroliths formed in the ureter and other organs evaluated through histopathology, there was no manifestation of urolithiasis, and there was no change in renal functions or obstruction of the urinary tract. Thus, the length of stay in the feedlot and the young age of the animals at slaughter may have been insufficient for total obstruction. These data corroborate the assumptions found in the literature that, despite being an important factor for the development of urolithiasis, the acidic urinary pH does not prevent the formation of uroliths if the urinary phosphorus levels are high [2]. The same authors reported that the determining factor for the formation of crystals that make up the urinary calculus is the high concentration of phosphorus in the urine, which has a high positive correlation with serum phosphorus, and that the acidic urinary pH does not prevent the formation of uroliths if the urinary phosphorus levels are high [2].

## 5. Conclusions

Supplementation with acidifying additives in lambs fed high-concentrate feedlot diets did not reduce feed intake, nor did it affect the performance and integrity of the ruminal and cecal epithelium. Most blood chemistry results were not significantly altered by treatments and remained within reference parameters. Acidifying additives were ineffective in preventing urolithiasis due to the presence of uroliths in the ureters of all animals and treatments, however, without obstruction of the urinary tract. Calcium chloride acidifies urinary pH and can be used instead of ammonium chloride at a concentration of 0.63% in the diet when this is the objective. However, future studies with other acidifying additives or different doses are recommended.

## Figures and Tables

**Figure 1 vetsci-10-00617-f001:**
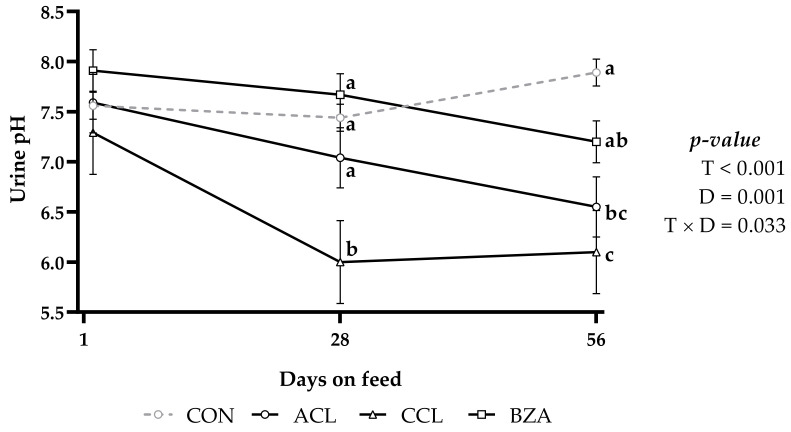
Urinary pH of lambs fed a high-concentrate diet with or without added additives and days on feed. CON: control; ACL: ammonium chloride; CCL: calcium chloride; BZA: benzoic acid. T: treatments; D: days on feed; T × D: treatments and days on feed interaction. Similar letters do not differ from each other by the least significant difference T-Fisher (LSD) test at the 5% probability level.

**Figure 2 vetsci-10-00617-f002:**
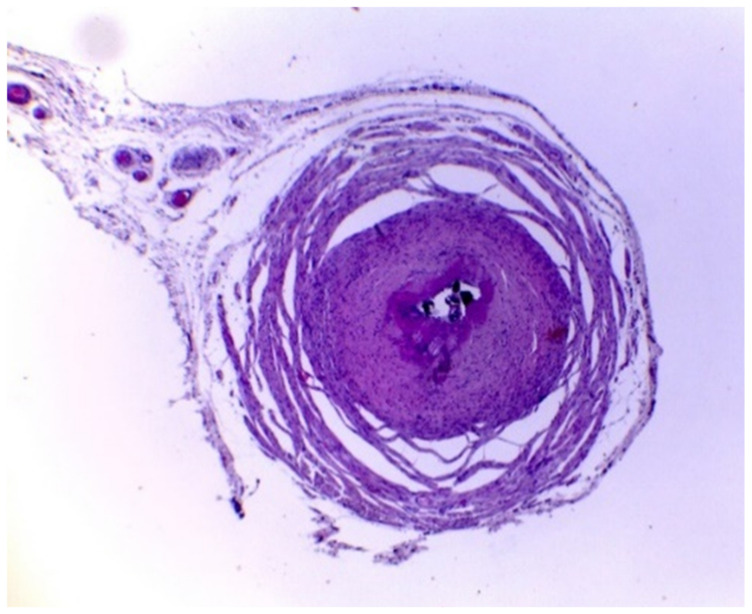
Ureter with the presence of stones and intense luminal stenosis in lamb. HE, obj. 4×. HE: hematoxylin and eosin; obj.4×: 4× magnification objective microscopic lens.

**Figure 3 vetsci-10-00617-f003:**
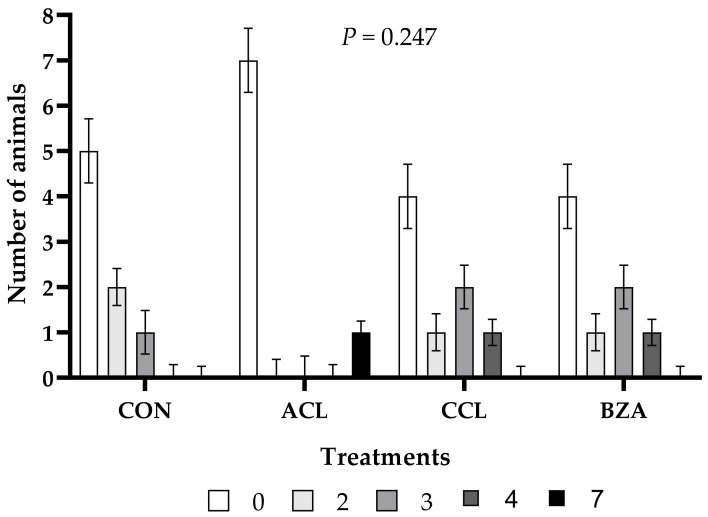
Rumenitis score of lambs fed a high-concentrate diet with or without added additives. CON: control; ACL: ammonium chloride; CCL: calcium chloride; BZA: benzoic acid. Similar letters do not differ from each other by the least significant difference T-Fisher (LSD) test at the 5% probability level.

**Figure 4 vetsci-10-00617-f004:**
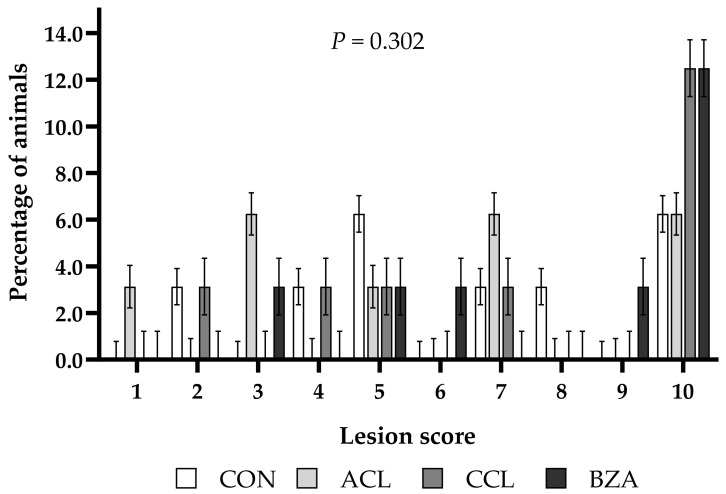
Cecal lesion score of lambs fed a high-concentrate diet with or without added additives. CON: control; ACL: ammonium chloride; CCL: calcium chloride; BZA: benzoic acid. Similar letters do not differ from each other by the least significant difference T-Fisher (LSD) test at the 5% probability level.

**Table 1 vetsci-10-00617-t001:** Composition and nutrient content of diets (% of dry matter basis, DM).

Ingredients, % of DM	Treatments ^1^			
	CON	ACL	CCL	BZA
Coastcross hay (*Cynodon* ssp.)	6.00	6.00	6.00	6.00
Ground corn	77.43	77.27	77.30	76.97
Soybean meal	13.00	13.00	13.00	13.00
Limestone	1.80	1.70	1.30	1.70
Urea	1.14	0.90	1.14	1.20
Calcium chloride ^2^	-	-	0.63	-
Ammonium chloride ^3^	-	0.50	-	-
Benzoic acid ^4^	-	-	-	0.50
Mineral premix ^5^	0.40	0.60	0.40	0.60
Sodium chloride	0.20	0.00	0.20	0.00
Selisseo ^6^	0.0013	0.0013	0.0013	0.0013
Monensin ^7^	0.004	0.004	0.004	0.004
Vitamin E ^8^	0.02	0.02	0.02	0.02
Total	100.00	100.00	100.00	100.00
**Nutrients, % of DM**				
Total digestible nutrients ^9^	81.69	81.55	81.58	81.28
Ether extract	2.40	2.41	2.54	2.49
Crude protein	17.24	17.39	17.22	17.36
Rumen degradable protein	10.90	10.22	10.90	11.04
Neutral detergent fiber	23.18	24.36	26.71	24.10
Calcium	0.77	0.76	0.77	0.76
Phosphorus	0.40	0.41	0.40	0.41
Sodium	0.10	0.09	0.10	0.09
Potassium	0.69	0.69	0.68	0.69
Chlorine	0.17	0.40	0.48	0.05
Sulfur	0.19	0.19	0.19	0.19
Diet cation–anionic balance ^10^, mEq/kg	+52.52	−17.10	−36.99	+81.52

^1^ CON: control without additive; ACL: ammonium chloride; CCL; calcium chloride; BZA: benzoic acid. ^2^ Kemin calcium chloride (Kemin, Verona, MO, USA): chlorine—49%, calcium—31%. ^3^ Ammonium chloride BASF. ^4^ Eastman Protural TM BA (Eastman, Madrid, Spain): benzoic acid—99.8%. ^5^ Coplasal Ovinos^®^ (Coplacana, Piracicaba, São Paulo, Brazil) calcium 155 g, phosphorus 83 g, magnesium—10 g, sulfur—35 g, sodium—120 g, copper—756 mg, magnesium—2110 mg, zinc—2800 mg, iodine—56 mg, cobalt—44 mg, selenium—14 mg, fluorine—250 mg. ^6^ Selisseo^®^ (Adisseo, Antony, Altos do Sena, France). ^7^ Rumenpac^®^ (Mcassab, São Paulo, Brazil): minimum monensin 200 mg/kg. ^8^ Microvit^®^ E Promix 50 (Adisseo, Antony, Altos do Sena, France): 500 IU/g, α-tocopherol. ^9^ Estimated according to Weiss et al. [14]. ^10^ Diet cation–anionic balance in mEq/kg = (% Na/0.0023 + % K/0.0039) − (% Cl/0.00355 + % S/0.00160).

**Table 2 vetsci-10-00617-t002:** Performance of lambs fed a high-concentrate diet with or without added additives.

Item ^1^	Treatments ^2^				SEM	*p*-Value
	CON	ACL	CCL	BZA		
Initial BW, kg	23.59	23.54	23.49	23.69	1.320	0.996
Final BW, kg	41.95	42.29	43.23	43.15	1.654	0.798
DMI, % BW	3.40	3.33	3.47	3.53	0.041	0.395
ADG, kg/day	0.333	0.341	0.357	0.351	0.010	0.793
FE (kg/kg)	0.276	0.290	0.285	0.281	0.009	0.902

^1^ BW: body weight; DMI: dry matter intake; ADG: average daily gain; FE: feed efficiency. ^2^ CON: control without additive; ACL: ammonium chloride; CCL; calcium chloride; BZA: benzoic acid.

**Table 3 vetsci-10-00617-t003:** Physicochemical parameters of urine of lambs fed a high-concentrate diet with or without added additives.

Item	Findings	Treatments ^1^				*p*-Value
		CON	ACL	CCL	BZA	
Color	Citrine yellow	8.89	12.22	14.44	13.33	0.001
	Gold yellow	8.89	1.11	8.89	7.78	
	Straw yellow	7.78	13.33	1.11	2.22	
Aspect	Clear	17.78	14.44	13.33	8.89	0.148
	Discreetly turbid	4.44	11.11	10.00	8.89	
	Turbid	3.33	1.11	1.11	5.56	
Density	<1.015	8.89	15.56	8.89	7.78	0.004
	1.015–1.040	6.67	10.00	8.89	2.22	
	>1.040	10.00	1.11	6.67	13.33	
Protein	Negative	2.22	7.78	4.44	1.11	0.006
	Trace	14.44	16.67	12.22	5.56	
	30–100 mg/dL	6.67	1.11	5.56	10.00	
	>100 mg/dL	2.22	1.11	2.22	6.67	
Blood	Absent	24.44	23.33	24.44	23.33	0.189
	Present	1.11	3.33	0.00	0.00	

^1^ CON: control without additive; ACL: ammonium chloride; CCL; calcium chloride; BZA: benzoic acid.

**Table 4 vetsci-10-00617-t004:** Morphometric parameters of ruminal papillae and cecal pH of lambs fed a high-concentrate diet with or without added additives.

Item	Treatments ^1^				SEM	*p*-Value ^2^
	CON	ACL	CCL	BZA		
Average area of ruminal papillae, cm^2^	0.277	0.224	0.282	0.259	0.014	0.465
Papillary surface absorption, %	90.67	91.10	90.94	90.13	0.398	0.783
Average number of ruminal papillae	35.23	45.57	35.96	36.38	1688	0.109
Absorptive surface area, cm^2^	10.29	10.79	10.59	8.91	0.468	0.302
Cecal pH	5.88 a	5.97 a	5.72 b	5.60 b	0.054	0.034

^1^ CON: control without additive; ACL: ammonium chloride; CCL; calcium chloride; BZA: benzoic acid. ^2^ Means without a common letter differ by the Student’s *t*-test (*p* < 0.05).

**Table 5 vetsci-10-00617-t005:** Blood erythrogram of lambs fed a high-concentrate diet with or without added additives and days on feed.

Item	Treatments (T) ^1^				SEM	Days (D)			SEM	*p*-Value ^2^		
	CON	ACL	CCL	BZA		1	28	56		T	D	T × D
Red blood cells, ×10^6^/µL	13.90	12.80	13.23	13.86	0.247	13.68	13.04	13.62	0.231	0.079	0.262	0.962
Hemoglobin, g/dL	13.10 ^a^	12.05 ^b^	12.68 ^b^	13.24 ^a^	0.163	13.11	12.5	12.7	0.142	0.043	0.278	0.957
Hematocrit, %	38.16	35.68	36.76	38.70	0.542	38.05	36.16	37.78	0.493	0.084	0.186	0.979
Mean corpuscular volume, fL	27.55	28.16	27.84	27.93	0.173	27.97	27.86	27.77	0.150	0.661	0.894	0.994
Mean corpuscular hemoglobin, pg	9.45	9.46	9.60	9.56	0.079	9.62 ^a^	9.60 ^a^	9.33 ^b^	0.076	0.536	0.013	0.983
Mean corpuscular hemoglobin concentration, %	34.32	33.66	34.54	34.25	0.184	34.46 ^a^	34.5 ^a^	33.62 ^b^	0.170	0.174	0.024	0.989

^1^ CON: control without additive; ACL: ammonium chloride; CCL; calcium chloride; BZA: benzoic acid. ^2^ T: treatments; D: days on feed; T × D: treatments and days on feed interaction. Means without a common letter differ by the Student’s *t*-test (*p* < 0.05).

**Table 6 vetsci-10-00617-t006:** Blood leukogram of lambs fed a high-concentrate diet with or without added additives and days on feed.

Item	Treatments (T) ^1^				SEM	Days (D)			SEM	*p*-Value ^2^		
	CON	ACL	CCL	BZA		1	28	56		T	D	T × D
Total leukocytes/µL	6925	7196	7350	7600	219.6	7431 ^ab^	6516 ^b^	7856 ^a^	190.2	0.743	0.044	0.911
Segmented neutrophils/µL	2608	2766	2964	3034	108.5	3270 ^a^	2612 ^b^	2646 ^b^	93.9	0.500	0.024	0.533
Lymphocytes/µL	3909	4016	3912	4233	181.2	3741 ^a^	3563 ^a^	4748 ^b^	162.3	0.881	0.007	0.911
Monocytes/µL	289	267	286	199	18.58	304	219	257	16.09	0.291	0.180	0.556
Eosinophils/µL	80	85	98	80	11.68	73	61	123	10.11	0.938	0.076	0.712
Basophils/µL	39	62	86	55	5.935	43	61	79	5.140	0.054	0.054	0.123

^1^ CON: control without additive; ACL: ammonium chloride; CCL; calcium chloride; BZA: benzoic acid. ^2^ T: treatments; D: days on feed; T × D: treatments and days on feed interaction. Means without a common letter differ by the Student’s *t*-test (*p* < 0.05).

**Table 7 vetsci-10-00617-t007:** Serum biochemistry, platelets, and plasma protein of lambs fed a high-concentrate diet with or without added additives and days on feed.

Item	Treatments (T) ^1^				SEM	Days (D)			SEM	*p*-Value ^2^		
	CON	ACL	CCL	BZA		1	28	56		T	D	T × D
Platelets	692.9	630.5	721.3	625.0	17.63	704.7	644.4	653.2	15.76	0.089	0.248	0.586
Plasma protein, g/dL	5.91	5.84	6.04	6.08	0.043	5.72 ^a^	5.85 ^a^	6.33 ^b^	0.037	0.168	<0.001	0.900
Urea, mg/dL	42.96	42.61	41.10	43.40	0.904	33.02 ^a^	45.69 ^b^	48.84 ^b^	0.783	0.820	<0.001	0.776
Creatinine, mg/dL	0.87	0.84	0.86	0.85	0.015	0.92 ^a^	0.76 ^b^	0.88 ^a^	0.013	0.816	<0.001	0.901
Phosphorus, mg/dL	8.93	8.38	8.67	8.70	0.194	7.59 ^a^	9.07 ^b^	9.34 ^b^	0.181	0.606	<0.001	0.438
Chloride, mmol/L	104.5	105.5	105.8	105.1	0.248	105.7	105.4	104.6	0.215	0.271	0.200	0.865
Sodium, mmol/L	135.8	136.4	137.2	137.5	0.364	132.6 ^a^	137.0 ^b^	140.6 ^c^	0.318	0.264	<0.001	0.792
Potassium, mmol/L	4.50	4.60	4.75	4.65	0.063	4.72	4.64	4.51	0.057	0.416	0.249	0.913

^1^ CON: control without additive; ACL: ammonium chloride; CCL: calcium chloride; BZA: benzoic acid. ^2^ T: treatments; D: days on feed; T × D: treatments and days on feed interaction. Means without a common letter differ by the Student’s *t*-test (*p* < 0.05).

## Data Availability

All data were published in this paper.

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
