# Peer review of "The Use of Additives to Prevent Urolithiasis in Lambs Fed Diets with a High Proportion of Concentrate"

_vetsci, 2023, doi:10.3390/vetsci10100617_

Round 1

Reviewer 1 Report

These results clearly summarize the main findings of your research. It should be noted that the study was conducted under specific conditions, and these results apply to those conditions. Additionally, providing more context, such as details about the care and feeding of the animals, environmental factors, and the duration of the experiment, could strengthen the study. The section where the findings are discussed is quite explanatory and consistent. It effectively explains how the findings address your research questions. References to similar studies in the literature can help contextualize your findings. Overall, there are very few grammar and writing errors. However, some sentences may be overly complex, and using simpler language could make it easier for readers to understand. Some sentences are a bit long and convoluted. Using shorter sentences can make the text flow more smoothly. The conclusions are presented clearly, and the main findings are summarized effectively. This helps readers quickly grasp the key results of the study. The suggested conclusions and recommendations for future research provide valuable insights. More emphasis could be placed on information about which organizations supported the study and how ethical approvals were obtained. The references are appropriately formatted as they appear in the text.

In conclusion, this study could be presented more effectively with careful editing and grammar checks. Summarizing the results and conclusions clearly will help readers better understand the main aspects of the study. Additionally, providing more context about the conditions under which the study was conducted and recommendations for future research could enhance the overall presentation.

Minor editing of English language required

Author Response

Reviewer 1

Comments and Suggestions for Authors

These results clearly summarize the main findings of your research. It should be noted that the study was conducted under specific conditions, and these results apply to those conditions. Additionally, providing more context, such as details about the care and feeding of the animals, environmental factors, and the duration of the experiment, could strengthen the study. The section where the findings are discussed is quite explanatory and consistent. It effectively explains how the findings address your research questions. References to similar studies in the literature can help contextualize your findings. Overall, there are very few grammar and writing errors. However, some sentences may be overly complex, and using simpler language could make it easier for readers to understand. Some sentences are a bit long and convoluted. Using shorter sentences can make the text flow more smoothly. The conclusions are presented clearly, and the main findings are summarized effectively. This helps readers quickly grasp the key results of the study. The suggested conclusions and recommendations for future research provide valuable insights. More emphasis could be placed on information about which organizations supported the study and how ethical approvals were obtained. The references are appropriately formatted as they appear in the text.

In conclusion, this study could be presented more effectively with careful editing and grammar checks. Summarizing the results and conclusions clearly will help readers better understand the main aspects of the study. Additionally, providing more context about the conditions under which the study was conducted and recommendations for future research could enhance the overall presentation.

AU: Thank you for the suggestion. The sentence requested in the Abstract was deleted. In a simple summary, we would like to maintain this sentence, as for lay readers, it can provide a context for the topic covered in this research (L.17-18). Other changes has been made on the manuscript following your suggestions.

Introduction.

The overall comprehensibility of the text is good; however, some sentences are excessively long and complex. Expressing these sentences in a more straightforward and simple manner can enhance reader understanding. Specifically, it is recommended to use simpler language when discussing ammonium chloride and other potential alternatives. For example, simplifying sentences by using phrases like "in the section where alternatives are examined" can make the text more accessible. Breaking down some sentences into shorter segments can facilitate the flow of the text. For instance, dividing a long sentence into two or more shorter sentences can make it more understandable.

AU: We appreciate the suggestions. We made some revisions and modifications throughout the introduction.

Materials and methods

AU: Table 1. We want to thank you for suggesting adding the abbreviation. However, we would like to keep the name in full as it concerns the ingredients.

  1. 129-133. The sentence can be expressed more clearly.

AU: Thanks for the suggestion. We made changes to be more clearly (L. 127-132).

L.138. Sampling and chemical analysis. Providing more details about the times at which chemical analyses and blood samples were collected can help us better understand how these procedures were carried out.

AU: It has been added to the manuscript.

Discussion

Your study examines the effects of the ACL, BZA and CCL on performance, but there are no references in the literature to previous studies comparing these items. This prevents the results from being placed in a wider context. Therefore, you may want to consider placing your study in a wider context with more references to the results of similar studies.

AU: We understand the reviewer concerning, however, most o the studies with the additives we had used in the present study are clinic studies, in this case, we keep the references about other species, which we understand it is valid to discuss our data.

  1. 356-360. I think it's inappropriate to give the result of a different kind of farm animal.

AU: We appreciate your suggestion. Indeed, pigs, as a biological model, may present a different response to BZA ingestion. However, in terms of comparison, making it clear that it is a different biological model to the one used in the present study, the comparison may be appropriate, given that there is little information on using BZA in sheep. Therefore, we adjusted the sentence, clarifying the biological model factor used in the study (L.357-362).

Conclusion

These findings clearly summarize the main outcomes of your research. It should be noted that your study was conducted under specific conditions, and these results are applicable to those conditions. Furthermore, adding more context, such as recommendations for future research or what these results did not illuminate, can further strengthen your conclusion section.

AU: Thanks. We made the adjustments suggested in the conclusion.

Reviewer 2 Report

The article is well written and delimited and has great scientific relevance. I have some suggestions: Insert the objective in the introduction; Table abbreviations must be defined in the captions; Authors should include a recommendation for future studies at the end of their discussion.  

Author Response

Thanks for the suggestion. The study's objective was inserted in the introduction (L.82-84). We inserted in Table 2 the legends of the abbreviations of the analyzed variables (L.200). Recommendations for future studies were included in the conclusion at the request of one of the reviewers of this manuscript (L. 471-472).

Reviewer 3 Report

This was a very interesting study which compared the use of additives to a high-concentrate diet on feed intake and growth performance along with various digestive and urinary system parameters, and urine and blood parameter. 

In general the manuscript was well presented, with specific feedback being provided in the attached PDF.

My major concern is with the histopathological results as it was not determined if the uroliths were present in the cohort of experimental animals used after the adaptation period but before treatment. Thus, the uroliths may have been present in all the animals before the experiment began. Thus, no conclusion regarding the prevention of urolithiasis can be made. I would suggest that the manuscript, therefore, needs reframing with the emphasis put on the impact of using various additives to a high concentrate diet on lamb growth, health and physiological parameters, with the objective of selecting "safe" additives that do not impact on feed intake or performance for use in a subsequent experiments that explores whether or not the additives are effective at preventing urolithiasis.

Minor suggestions to the use of the English language and paragraph structure have been annotated in the attached PDF.

Author Response

Reviewer 3

Comments and Suggestions for Authors

This was a very interesting study which compared the use of additives to a high-concentrate diet on feed intake and growth performance along with various digestive and urinary system parameters, and urine and blood parameter.

In general, the manuscript was well presented, with specific feedback being provided in the attached PDF.

My major concern is with the histopathological results as it was not determined if the uroliths were present in the cohort of experimental animals used after the adaptation period but before treatment. Thus, the uroliths may have been present in all the animals before the experiment began. Thus, no conclusion regarding the prevention of urolithiasis can be made. I would suggest that the manuscript, therefore, needs reframing with the emphasis put on the impact of using various additives to a high concentrate diet on lamb growth, health and physiological parameters, with the objective of selecting "safe" additives that do not impact on feed intake or performance for use in a subsequent experiment that explores whether or not the additives are effective at preventing urolithiasis.

Comments on the Quality of English Language

Minor suggestions to the use of the English language and paragraph structure have been annotated in the attached PDF.

AU: Thanks for the comment. All lambs came from a farm without urolithiasis reported for over three years. On the farm, lambs from 10 days of age were supplemented via creep-feeding. The supplement was 6% powdered milk, 2% mineral core, 2% limestone, 70% ground corn, and 20% soybean meal. In addition to supplementation in creep feeding, Tifton-85 hay was offered to both the lambs and their dams. The lambs were acquired from the farm very young (±50 days of age) and remained on the feedlot for 56 days. Several studies have reported cases of urolithiasis in lambs over three months of age and with a higher incidence in castrated males. We believe that the lambs in the present study were too young to have urolithiasis when they entered the experimental phase and were non-castrated, further reducing the incidence of the disease.

It would make more sense to use the chemical formula here as opposed to the treatment group label. Otherwise make it very clear that this is a treatment group label.

AU: Thanks for the suggestion. We would like to keep the abbreviations since some chemical formulas of treatments are large. On the other hand, some readers may be unable to be familiar with chemical formulas, causing possible reading difficulties.

Introduction.

L.63-64. “its reversal is hardly thriving”. It is not clear what the intended mean is here.

AU: Thanks for the comment. We rewrote the sentence to clarify (L.61-62).

There needs to be a sentence here saying something along the lines of "In sheep, one of the most common types of urolith is composed of calcium carbonate"

AU: Thanks for the comment. We rewrote the sentence to clarify (L.61-62).

“The formation of crystals occurs” ….

AU: It is calcium carbonate.

What does NRC stand for?

AU: Is the American System of Nutritional Requirements for Small Ruminants (Nutrient Requirements of Small Ruminants Sheep, Goats, Cervids, and New World Camelids). The abbreviation NRC is well-known in animal nutrition. We would like to keep the abbreviation in the text, so as not to make the sentence long.

Some more explanation as to why benzoic acid is required as it is not typically used to create anionic diets. For example, after this sentence it could be stated that "Unlike calcium chloride, benzoic acid does not increase the anionic content of the animals diet, instead it is metabolized by the liver to form hippuric acid which is excreted in the urine lower it's pH." Appropriate references would need to be included.

AU: Thanks for the suggestion. We put the recommended sentence in the example and added the references (L.81-83).

“Thirty-two male lambs, non-castrated, crossed Dorper × Santa Inês, with initial BW 97 of 23 ± 5 kg and age of 50 ± 5 days were housed in individual pens (2 × 1.25 m), with ad 98 libitum access to feed and water”.

Was the food offering remove overnight?

AU: The animal had access to feed throughout the day. The feed was not removed during the night.

Suggest rewriting "The urea and creatinine concentrations in these blood samples were determined using commercial kits (Labtest, Lagoa Santa, Minas Gerais, Brazil) through the following methods: enzymatic-colorimetric and kinetic-colorimetric, respectively, with readings performed on a spectrophotometer."

AU: Thanks for the suggestion. We rewrite as recommended (L.145-149).

“The urine collections were carried out at 7:00 am before feeding”….

It has been stated that feeding was ad libitum. Was the food offered removed overnight? This needs to be clarified.

AU: Feed was not removed at night and the animal had feed available throughout the day.

The animal was given a daily amount of feed so that it had 10% left over. New feed was placed in the trough at 7 o'clock in the morning. Urine samples were collected prior to this dietary management.